# Histological and immunohistochemical characterization of the porcine ocular surface

Mario Crespo-Moral[1☯], Laura García-Posadas[1☯], Antonio López-García[1,2], Yolanda Diebold[1,2]*

1 Ocular Surface Group, IOBA – University of Valladolid, Valladolid, Spain, 2 Biomedical Research Networking Center on Bioengineering, Biomaterials and Nanomedicine (CIBER-BBN), Valladolid, Spain

☯ These authors contributed equally to this work.
* yol@ioba.med.uva.es

## Abstract

The ocular surface of the white domestic pig (*Sus scrofa domestica*) is used as a helpful model of the human ocular surface; however, a complete histological description has yet to be published. In this work, we studied porcine eyeballs with intact eyelids to describe and characterize the different structures that form the ocular surface, including the cornea and conjunctiva that covers the bulbar sclera, tarsi, and the nictitating membrane. We determined the distribution of goblet cells of different types over the conjunctiva and analyzed the conjunctival-associated lymphoid tissue (CALT). Porcine eyeballs were obtained from a local slaughterhouse, fixed, processed, and embedded in paraffin blocks. Tissue sections (4 μm) were stained with hematoxylin/eosin, Alcian blue/Periodic Acid Schiff, and Giemsa. Slides were also stained with lectins from *Arachis hypogaea* (PNA) and *Helix pomatia* (HPA) agglutinins and immunostained with rabbit anti-CD3. We found that the porcine cornea was composed of 6–8 epithelial cell layers, stroma, Descemet's membrane, and an endothelial monolayer. The total corneal thickness was 1131.0±87.5 μm (mean±standard error of the mean) in the center and increased to 1496.9±138.2 μm at the limbus. The goblet cell density was 71.25±12.29 cells/mm, ranging from the highest density (113.04±37.21 cells/mm) in the lower palpebral conjunctiva to the lowest density (12.69±4.29 cells/mm) in the bulbar conjunctiva. The CALT was distributed in the form of intraepithelial lymphocytes and subepithelial diffuse lymphoid tissue. Lenticular-shaped lymphoid follicles, about 8 per histological section, were also present within the conjunctival areas. In conclusion, we demonstrated that the analyzed porcine ocular structures are similar to those of humans, confirming the potential usefulness of pig eyes to study ocular surface physiology and pathophysiology.

## Introduction

The ocular surface is the interface between the eye and the environment. Classically, it is comprised of the corneal, limbal, and conjunctival epithelia and the tear film [1]. However,

**Data Availability Statement:** All data files are available from the Figshare database (https:// figshare.com/articles/Histological_and_ immunohistochemical_characterization_of_the_

porcine_ocular_surface_Raw_data_xlsx/
11417130).

**Funding:** YD, MAT2013-47501-C2-1-R (Spanish Ministry of Economy and Competitiveness and European Regional Development Fund) http://www.mineco.gob.es/ and https://ec.europa.eu/regional_policy/en/funding/erdf/ YD and LGP, RTI2018-094071-B-C21 (Spanish Ministry of Science, Innovation and Universities and European Regional Development Fund) http://www.ciencia.gob.es/portal/site/MICINN/ and https://ec.europa.eu/regional_policy/en/funding/erdf/ LGP, Postdoctoral contrats 2017 call (University of Valladolid) http://www.uva.es/export/sites/uva/MCM, Regional JCyL Scholarship/European Social Fund Program ORDEN EDU/128/2015 (Regional JCyL and European Social Fund) https://www.jcyl.es/ and https://ec.europa.eu/esf/home.jsp The funders had no role in study design, data collection and analysis, decision to publish, or preparation of the manuscript.

**Competing interests:** The authors have declared that no competing interests exist.

the concept of the ocular surface has evolved in the last 20 years to a more complex pathophysiological functional unit [2]. In this initial report, we confined our analysis to the components of the traditionally recognized ocular surface and the Meibomian gland.

Due to the contact with the external environment, the lacrimal functional unit, and specifically the ocular surface, multiple defensive mechanisms exist. The corneal epithelium forms a tight barrier that impedes the entrance of pathogens. However, because the cornea must be transparent to allow the transmission of light, it has no blood vessels and depends on other tissues to support it. The corneal epithelium is renewed by epithelial stem cells located in the Palisades of Vogt, which are radially-oriented fibrovascular ridges present in the limbus, the area between the cornea and the conjunctiva [3]. The limbal tissue transitions into the conjunctiva, a mucosal tissue that, unlike the cornea, is highly vascularized and displays a strong reaction against antigens and infections without compromising the maintenance and/or recovery of ocular surface homeostasis [4]. Indeed, the conjunctiva is the major supporting tissue of the ocular surface. One of its main functions is attributed to the presence of mucin-secreting goblet cells that play a role in protecting the ocular surface. These mucins, along with the water and electrolytes secreted by the lacrimal gland and the lipids produced by the Meibomian glands, form the tear film [5]. Alterations in the function of goblet cells lead to changes in tear composition that can result in different pathologies [6,7].

The conjunctiva also possesses specific lymphoid components belonging to the mucosa-associated lymphoid tissue (MALT) that can locally initiate and regulate immune responses [8,9]. In the conjunctiva, the MALT is present as the conjunctival-associated lymphoid tissue (CALT) and consists of a diffuse layer of lympho-epithelium and lymphoid follicles composed of B and T lymphocytes, macrophages, and dendritic cells. The CALT functions as the efferent and afferent arms of the conjunctival immune system [10]. The lymphoid follicles are overlain by a specialized follicle-associated epithelium (FAE) that is thinner than the regular conjunctival epithelium, has a fragmented basal lamina, lacks goblet cells, and contains M-cells that transport antigens [8]. Another typical feature of CALT is the presence of high endothelial venules (HEVs) associated with the follicles and that facilitate the migration of lymphocytes to these areas of the conjunctiva.

Experiments conducted to study the ocular surface can be performed in different *in vitro*, *ex vivo*, or *in vivo* models. Several cell lines have been widely used to study the corneal [11–14] and conjunctival [13,15] epithelia. However, the use of cell lines is being questioned due to frequent problems of misidentification and cross-contamination [16,17]. For that reason, primary cultures are emerging as the best way to study cell physiology *in vitro*. In addition, *ex vivo* models are an excellent tool to deepen the knowledge of physiological features without the disadvantages of *in vivo* research. Unfortunately, the availability of human tissue to perform *in vitro* or *ex vivo* studies is highly limited, a situation that constrains this type of experimentation. Therefore, the use of animal tissue is necessary. Humans and pigs share similar anatomic and physiologic characteristics that make pigs useful as experimental models in biomedical research [18,19]. Several studies have reported the characteristics of pig eyes, including parameters of the whole eyeball [20,21], retina [22,23], cornea [24], limbus [25], and the lacrimal gland [26]. However as far as we know, a thorough description of the conjunctiva of the pig eye has not yet been reported. Thus, the purpose of this study is to provide a detailed description of the pig ocular surface with special emphasis on the conjunctiva. Our goal was to determine the suitability of porcine conjunctiva as a model to advance knowledge of the human ocular surface and associated diseases.

## Materials and methods

### Porcine eyes

All experiments were conducted following the Association for Research in Ophthalmology Statement for the Use of Animals in Ophthalmic and Vision Research (https://www.arvo.org/About/policies/statement-for-the-use-of-animals-in-ophthalmic-and-vision-research/) and were approved by the Ethics Committee of the University of Valladolid. Eyeballs with eyelids (n = 3) were obtained from white domestic pigs (*Sus scrofa domestica*) donated by the local slaughterhouse Justino Gutiérrez SL (Laguna de Duero, Valladolid, Spain). The animals were between 6 and 8 months of age (pre-adults), and weighed 120–150 kg. The chief veterinarian of the slaughterhouse performed the exenterations immediately after each pig was killed, and the exenterated tissues were immediately placed in 4% buffered paraformaldehyde. Afterwards, tissues were transported to the laboratory where they were maintained in the fixative solution for ten days.

### Tissue processing

All the adjacent muscle, fat, and connective tissue were dissected and removed to finally process the fixed eyeball and eyelids in a tissue processor (Leica Biosystems ASP300, Nussloch, Germany) for 16 hours. Paraffin tissue blocks were prepared and 4-μm–thick sections were obtained using a soft tissue microtome (Microm, Walldorf, Germany).

### Histological staining and light microscopy analysis

Ocular sections were deparaffinized with xylene (Applichem Panreac, Barcelona, Spain) and rehydrated through a decreasing gradient of ethanol (Applichem Panreac) followed by de-ionized water. Then, the sections were stained with hematoxylin/eosin (H/E), Alcian blue/periodic acid Schiff (AB/PAS), or Giemsa.

H/E staining was used to identify and describe the different tissues and structures within them. Rehydrated slides were rinsed for 1 min in distilled water, immersed in Mayer's Hematoxylin (Millipore, Billerica, MA, USA) for 5 min, and rinsed in running tap water for 10 min. Then, they were rinsed in 80% ethanol for 1 min and immersed in eosin (Fluka, Buchs, Switzerland) for 5 min. Finally, the slides were dehydrated, cleared, and mounted with cover slips.

AB/PAS-stained slides were used to identify and count goblet cells. Slides were rinsed in distilled water, immersed in 3% acetic acid for 3 min followed by Alcian blue solution (pH 2.5; Sigma Aldrich, St. Louis, MO, USA) for 15 min. After that, the slides were immersed in 0.5% periodic acid for 5 min, rinsed with distilled water, and placed in Schiff's solution (Millipore) for 15 min. The slides were then rinsed in running tap water for 10 min, counterstained with Mayer´s Hematoxylin (Millipore) for 5 min, and finally rinsed, dehydrated, cleared, and mounted with cover slips. Acidic (blue), neutral (pink), and mixed (purple) goblet cells were counted in each conjunctival area. Goblet cell density (GCD) was calculated by counting the number of goblet cells in a section and dividing that number by the length of the section. At least three different sections of ≥500 μm were counted for each conjunctival region. Mean values for the number of cells counted independently by two researchers were then calculated.

Giemsa-stained sections were used to analyze the presence of inflammatory cells and characterize the CALT in porcine conjunctiva. Briefly, rehydrated slides were placed in 20% Giemsa solution (Merck, Darmstadt, Germany) for 1 h and then rinsed in distilled water. The sections were then differentiated with 0.5% acetic acid, dehydrated rapidly, cleared, and mounted.

## Lectin staining

To detect glycoconjugates produced by conjunctival goblet cells, we used lectins from *Arachis hypogaea* agglutinin (PNA) and from *Helix pomatia* agglutinin (HPA), which bind β-D-galactose-1→3-D-N-acetyl-galactosamine and α-N-acetyl-α-D-galactosamine residues, respectively. Ocular sections were deparaffinized and rehydrated as previously described. The slides were washed three times with phosphate-buffered saline (PBS) and then fluorescein isothiocyanate (FITC)-conjugated PNA (Sigma Aldrich, L7381, Lot 056k4006) and tetramethylrhodamine (TRITC)-conjugated HPA (Sigma Aldrich, L1261, Lot 091k3793) at 1:500 dilution were added and incubated for 40 min at room temperature. After that, the slides were washed 3 times with PBS to remove the excess lectins. Cell nuclei were counterstained with Hoechst 33342 dye (Sigma Aldrich, B2261) at 1:1000 dilution. The preparations were viewed under an epifluorescence microscope (Leica DMI 6000B; Leica Microsystems, Wetzlar, Germany).

## T lymphocyte immunodetection

To determine T lymphocyte locations within the porcine conjunctiva, ocular sections were stained with anti-CD3 antibody. Prior to immunodetection, enzymatic antigen retrieval was performed in the sections with 0.05% trypsin. Then, sections were immunostained with rabbit monoclonal anti-CD3 (Abcam, ab16669, Cambridge, UK) at a 1:200 dilution for 18 h at 4˚ C. After that, the sections were incubated with polyclonal donkey anti-rabbit Alexa Fluor® 488 (Thermo Fisher Scientific, Waltham, MA, USA) secondary antibody (1:200 dilution) for 1 h at room temperature. Cell nuclei were counterstained with Hoechst 33342 at a 1:1000 dilution. Negative controls included the omission of primary antibody and positive controls included human tonsil. Slides were observed under the Leica DMI 6000B epifluorescence microscope and representative micrographs were taken at different magnifications.

## Automated image acquisition

AB/PAS-stained slides were analyzed using the Automated Cellular Imaging System III (ACIS III; Dako, Glostrup, Denmark). The system automatically acquired digital images of the glass slides using the ACIS scanner at low magnification (x10). Using ACIS III functionalities, we measured the thickness of the cornea in three different regions: at approximately the corneal apex, in the intermediate between the apex and the limbus, and at the periphery near the limbus.

## Data presentation and statistical analysis

Data were presented as means ± standard errors of the mean. Statistical differences in corneal thickness were analyzed with Student's *t*-test. Statistical differences in GCD were analyzed by one-way analysis of variance. Then, pairwise comparisons were performed with Tukey's test. Results were considered significantly different at $p \leq 0.05$. Statistical analyses were conducted using the Statistical Package for the Social Sciences software (SPSS).

# Results

## Macroscopic description of the pig eyeball and eyelids

The macroscopic anatomy of the pig eyeball and eyelids is similar to that of humans, although the pig eye has a bigger iris and a thicker cornea. Regarding the eyelids, the main and most obvious difference between pigs and humans was the presence of the nictitating membrane, also known as the third eyelid, in the pig eyes (Fig 1). The nictitating membrane was situated at the medial angle of the eye and contained cartilage that provides structural support as the lid

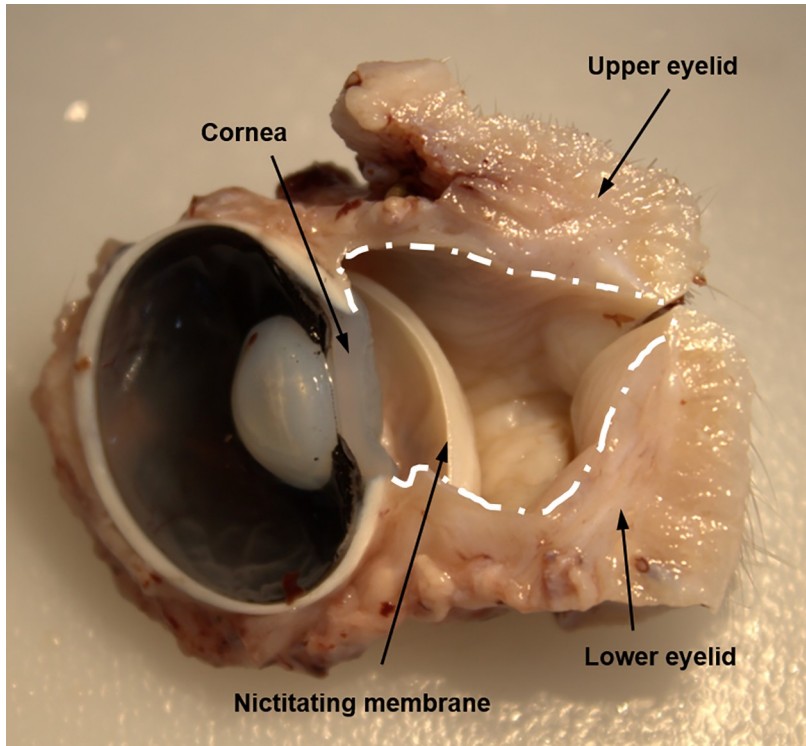

**Fig 1. Macroscopic photograph of porcine eye.** Transverse section of a fixed pig eyeball with the eyelids. The white line marks the limits of the conjunctiva.

moves horizontally across the eyeball. In contrast, the upper and lower eyelids, which were similar in size and shape to one another, move vertically across the eyeball.

## The porcine cornea

Based on histological analysis, the thickness of central cornea, 1,131.0 ± 56.3 μm, was slightly thinner than the intermediate area, 1,215.2 ± 32.9 μm (p < 0.187), that is located between the central cornea and the limbus. The limbal cornea was the thickest, 1,496.9 ± 60.8 μm (p < 0.001 vs central cornea, p < 0.001 vs intermediate cornea). The porcine cornea was composed of four layers: epithelium, stroma, Descemet's membrane, and endothelium (from outermost to innermost, Fig 2). The epithelium was stratified and composed of 6 to 8 layers, distributed as 2–3 layers of superficial stratified squamous cells, 2–3 layers of intermediate wing cells, and 2 layers of basal cells. The basal cells rested on a continuous basement membrane. Bowman's layer was not evident in the porcine cornea.

The stroma was the thickest layer of the cornea. It was composed of a structured collagen fiber matrix in which elongated keratocytes were embedded. Attached to the innermost part of the stroma was Descemet's membrane, and underneath it there is a single layer of endothelial cells.

## The porcine limbus

The limbus is the transition zone between the cornea and the conjunctiva. The limbal epithelium had a special anatomical conformation with the palisades of Vogt (Fig 3), where limbal stem cells are found [27]. In the pig eye, the limbal epithelium was composed of 12 layers of

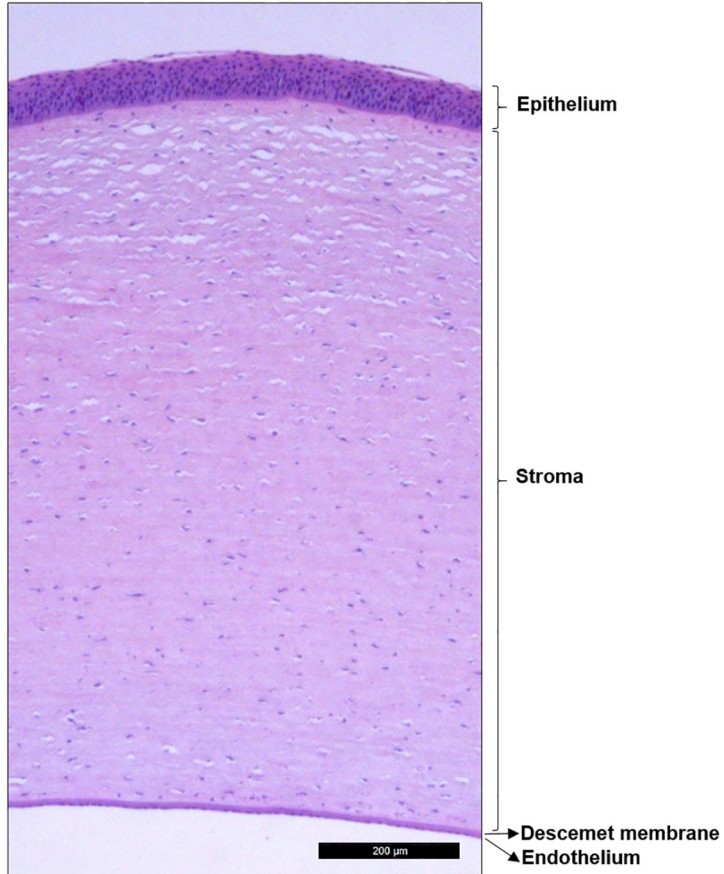

**Fig 2. Histological analysis of the porcine cornea.** Tissue section stained with hematoxylin-eosin showing the four layers of the cornea.

epithelial cells: 3 layers of flattened superficial squamous cells, 6 layers of intermediate wing cells, and 3 layers of basal cells.

## The porcine conjunctiva

The porcine conjunctiva was composed of a stratified epithelium and the substantia propria or stroma. The conjunctiva had three main anatomical zones: palpebral conjunctiva, bulbar conjunctiva, and fornix (*cul-de-sac*) (Fig 4). The palpebral conjunctiva lined the posterior surface of the eyelids. It consisted of the marginal conjunctiva at the edges of the lids and the tarsal conjunctiva. The bulbar conjunctiva was attached to the sclera. Finally, the fornix connected the palpebral conjunctiva with the bulbar conjunctiva.

The appearance of the porcine conjunctiva varied in the different regions, presenting diverse characteristics and a variable number of epithelial cell layers. At the marginal conjunctiva between the tarsal and palpebral regions, several infoldings of the epithelium (crypts) were present (Fig 5A). In this area, the conjunctiva consisted of 6 epithelial cell layers, including 2 superficial layers of squamous cells and 4 deeper layers of cuboidal cells (Fig 5A). In the tarsal conjunctiva, the crypts disappeared, and the epithelium consisted of 8 layers (Fig 5B). In the fornix, the histological staining revealed 6 epithelial cell layers (Fig 5C), whereas in the bulbar conjunctiva there were only 4 layers, but the cells were larger so that the total epithelial

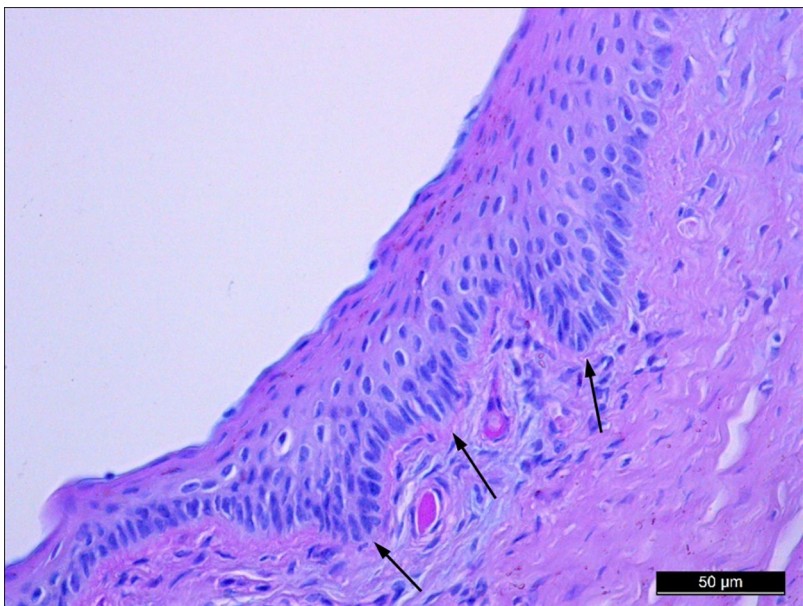

**Fig 3. Histological analysis of the porcine limbus.** Tissue section of porcine limbus stained with Alcian blue/periodic acid Schiff showing the characteristic palisades of Vogt (arrows) where limbal epithelial stem cells reside.

thickness was maintained (Fig 5D). As described in a following section, the densities of goblet cells varied in these regions.

The organization of the conjunctival stroma was dissimilar to that of the cornea. It was composed of loose connective tissue that included a superficial lymphoid layer and a deeper

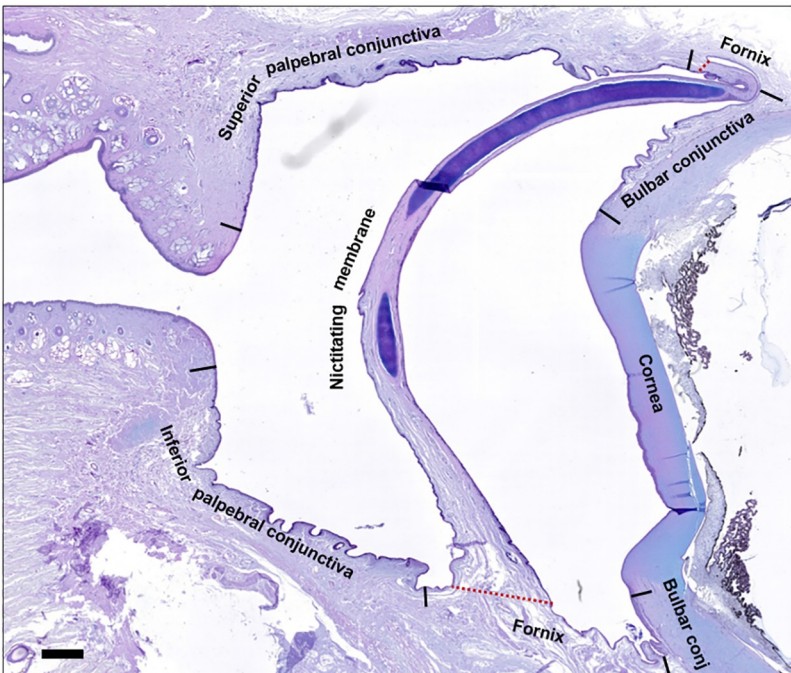

**Fig 4. Histological analysis of the porcine ocular surface.** Low magnification tissue section of porcine anterior ocular surface stained with Alcian blue/periodic acid Schiff. Bar = 1 mm.

                                                                                              

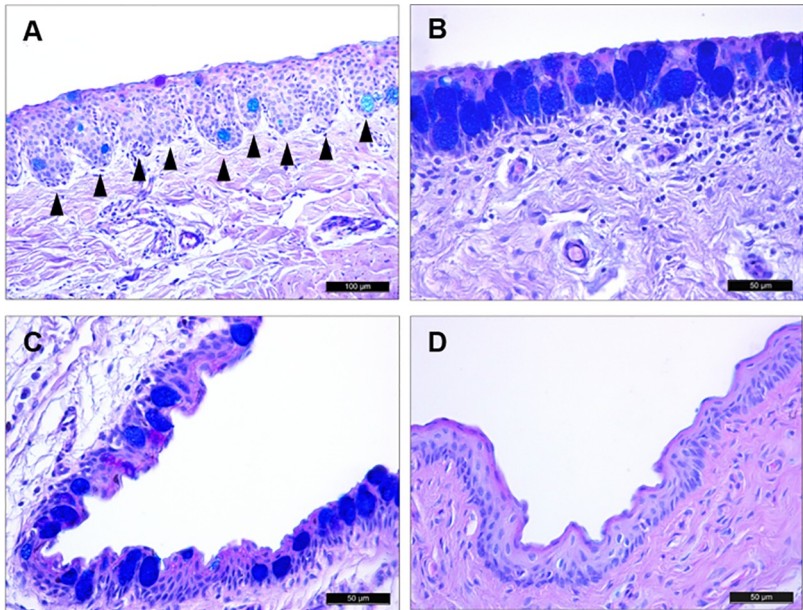

**Fig 5. Histological analysis of porcine conjunctiva.** Tissue sections of pig conjunctiva stained with AB/PAS. (A) The marginal conjunctiva between the tarsal and palpebral surfaces was covered by a stratified squamous epithelium and deeper cuboidal epithelial cells. Epithelial downgrowths into the stroma appeared as crypts (arrowheads). (B) The tarsal conjunctiva had a large number of goblet cells containing acidic glycoconjugates. (C) Conjunctiva in the fornix. (D) The bulbar conjunctiva had 4 epithelial cell layers and very few goblet cells.

fibrous layer with a large number of fibroblasts. In addition, there was a significant presence of blood vessels and immune cells (see *Location and characteristics of lymphoid tissue in porcine conjunctiva* section below).

The characteristics of the conjunctival epithelium in the nictitating membrane varied depending on the area (Fig 6A). In the zones over the cartilage, the epithelium consisted of 4 to 6 layers of stratified squamous cells (Fig 6B). At the edges, the nictitating membrane was composed of 12 epithelial cell layers (Fig 6C), whereas in the center it had only 8 layers (Fig 6D).

**Goblet cell characteristics and distribution in porcine conjunctiva.** Numerous goblet cells were present in the porcine conjunctival epithelium. The large, rounded cells were filled with glycoconjugate (mucin) granules. The glycoconjugate components of these cells differed from cell to cell and were differentiated by AB/PAS staining (Fig 7A). The acidic glycoconjugates were stained blue by AB, and the neutral glycoconjugates were stained pink by PAS. Some goblet cells contained a mixture of blue- and pink-stained glycoconjugates, appearing as purple granules. In the porcine conjunctival epithelium, 7.59 ± 0.94% of the total goblet cells were neutral, 5.38 ± 1.65% were acidic, and 87.02 ± 1.36% were mixed.

The distribution of goblet cells varied along the conjunctival epithelium topography, resulting in variations of GCDs in the different eyelids. GCD was 84.07 ± 9.59 cells/mm in the upper eyelid and upper fornix, 87.51 ± 9.16 cells/mm in the lower eyelid and lower fornix, and 71.44 ± 4.86 cells/mm in the anterior and posterior surfaces of the nictitating membrane. In addition to the conjunctival epithelium, goblet cells were also present in stromal structures similar to human pseudogland of Henle These crypts were sparsely distributed in both the superior and inferior conjunctival fornices, but they were absent in other areas (Fig 7B).

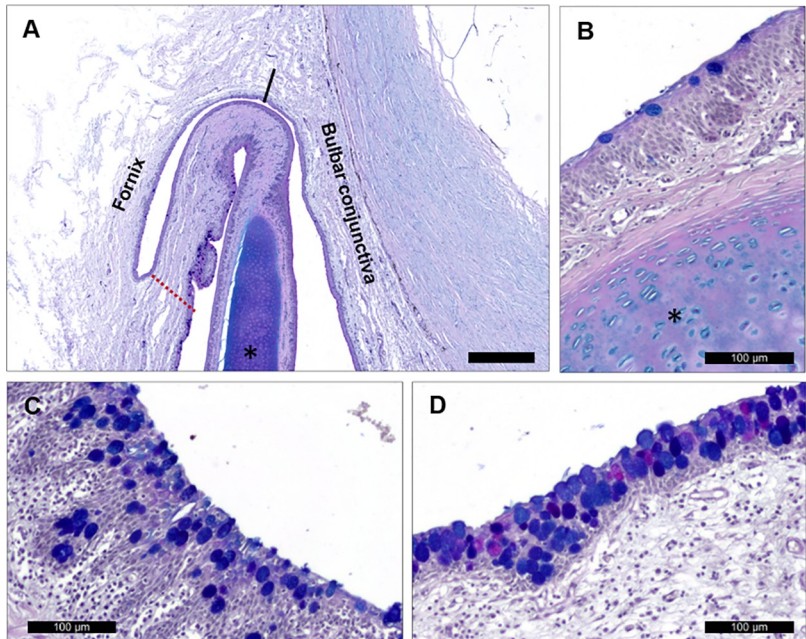

**Fig 6. Porcine nictitating membrane.** (A) Low magnification micrograph of the nictitating membrane union with upper eyelid. Bar = 500 μm. (B) Goblet cells in the conjunctival epithelium over the nictitating membrane cartilage. (C) Conjunctival edge between nictitating membrane and lower eyelid. (D) Conjunctiva over the center of the nictitating membrane anterior surface showing great abundance of goblet cells containing acidic, neutral, or both types of glycoconjugates.

For further analysis of goblet cell distribution, we divided the conjunctiva into 7 different areas: (1) upper palpebra, (2) upper fornix, (3) bulbar, (4) lower palpebra, (5) lower fornix, (6) anterior nictitating membrane, and 7) posterior nictitating membrane. The bulbar conjunctiva had the lowest GCD, 12.69 ± 4.29 cells/mm (Fig 8). The highest GCDs were in the upper and lower palpebral conjunctivas, 103.20 ± 15.19 cells/mm and 113.04 ± 5.76 cells/mm respectively, and both were greater than each of the other areas (p < 0.05 for all comparisons).

We also analyzed goblet cell contents based on lectin staining. Porcine conjunctival goblet cells stained with HPA and PNA lectins, indicating the presence of α-N-acetyl-α-D-galactosamine and β-D-galactose-1→3-D-N-acetyl-galactosamine, respectively (Fig 9).

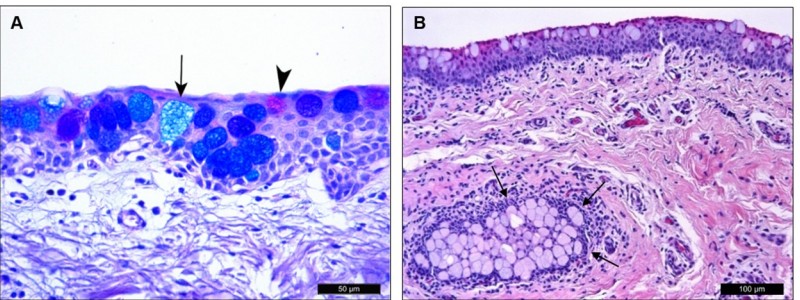

**Fig 7. Porcine conjunctival goblet cells.** (A) The different types of goblet cells can be distinguished with AB/PAS staining. Acidic glycoconjugates were stained blue (arrow) by AB, and neutral glycoconjugates were pink (arrowhead) by PAS. Most goblet cells have both types of glycoconjugate granules and appear as dark blue or purple color. (B) H/E staining showed a pseudogland of Henle (arrows) formed by a group of goblet cells embedded within the conjunctival stroma.

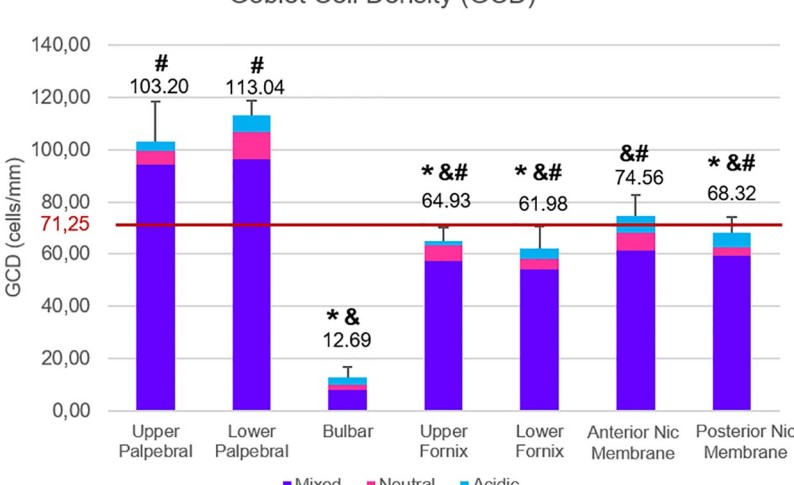

**Fig 8. Goblet cell density (GCD) in the different conjunctival regions.** The bar at 71.25 cells/mm represents the mean GCD for the whole conjunctiva. $p \leq 0.005$ for *, vs upper palpebral; &, vs lower palpebral; #, vs bulbar.

**Location and characteristics of lymphoid tissue in porcine conjunctiva.** Whereas no lymphoid cells were observed in the cornea, light microscopy revealed the presence of CALT in the porcine conjunctiva (Fig 10A). The CALT consisted of intraepithelial lymphocytes, lymphoid follicles, and subepithelial diffuse lymphoid tissue that followed a regional distribution in the conjunctival tissues. In addition to the organized follicles located mainly in the palpebral area, diffuse lymphoid tissue was present throughout the pig conjunctiva.

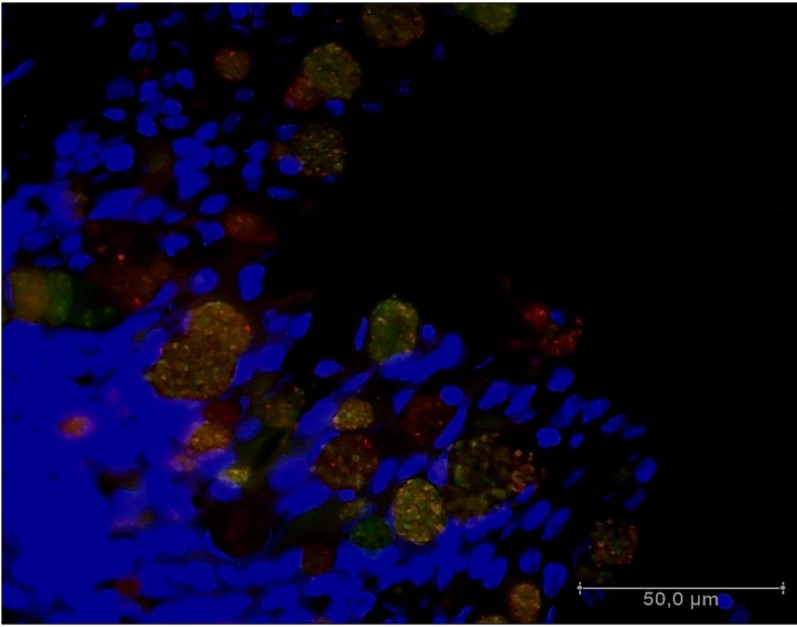

**Fig 9. Lectin binding in porcine conjunctiva.** Lectins from *Arachis hypogaea* agglutinin (PNA, green) and *Helix pomatia* agglutinin (HPA, red) bound to porcine lower palpebral conjunctival goblet cells.

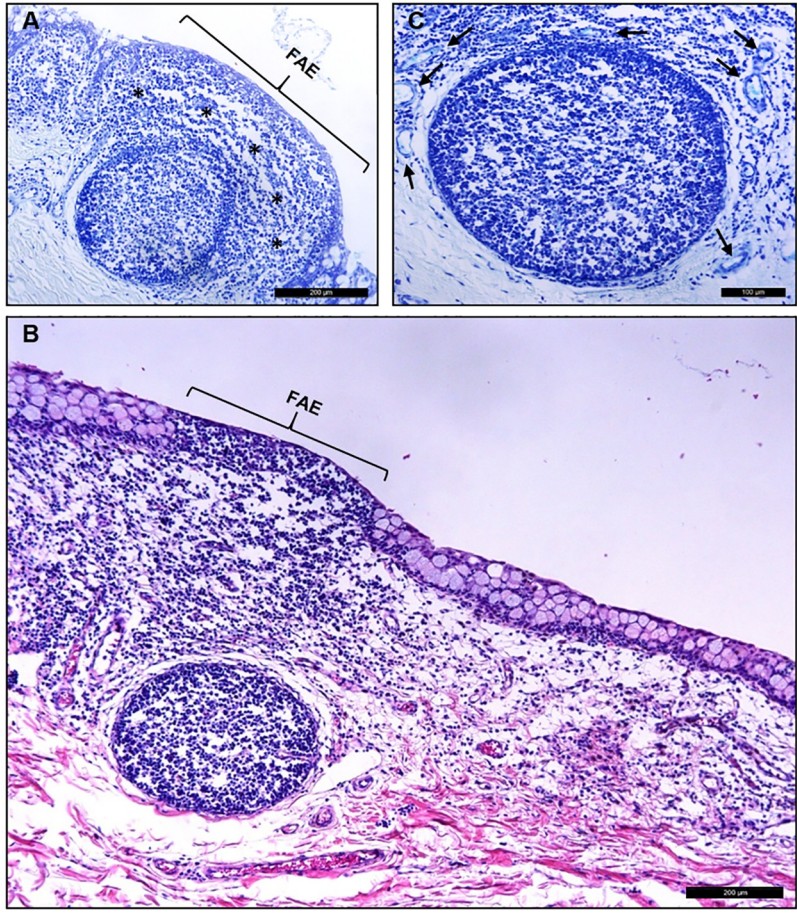

**Fig 10. CALT in porcine conjunctiva.** (A) The palpebral conjunctiva contained diffuse lymphoid tissue (asterisks) and a follicle. (B) Conjunctival section stained with H/E, showing the presence of CALT in the porcine conjunctiva. A diffuse lymphoid layer was present underneath the epithelium. In this representative image, there was a well-developed follicle in the stroma, and the characteristics of the follicle-associated epithelium (FAE) were evident. FAE, follicle-associated epithelium. (C) Tissue section of palpebral conjunctiva stained with Giemsa showing the presence of a well-defined follicle and HEVs (arrows).

The distribution of this lymphoid tissue was not homogeneous within the entire conjunctiva, with some areas having larger accumulations of immune cells than others. Usually, in the areas with large accumulations, lymphoid follicles were present. There were 8.67 ± 2.96 follicles in the analyzed sections of the whole eye. The follicles were lenticular in shape, with a mean large diameter of 188.9 ± 31.2 μm and a short diameter of 161.3 ± 26.2 μm. The superior and inferior palpebral conjunctivas had the greatest abundance of lymphoid follicles. In contrast, they were absent in the bulbar conjunctiva.

In the areas of the greatest lymphoid infiltration and follicle presence, goblet cells were scarce or even absent. In these areas, the basement membrane was usually discontinuous, and the epithelium was thinner, showing the typical characteristics of the FAE (Fig 10A and 10B). In addition, wherever the lymphoid tissue was present, an abundance of HEVs were evident (Fig 10C). The HEVs had thicker walls than normal venules, and the endothelial cells were more cuboidal in shape.

Immunostaining of CD3+ cells enabled detection of T lymphocytes in the conjunctiva. CD3+ T cells were present around and within the follicles, in the diffuse lymphoid tissue, and

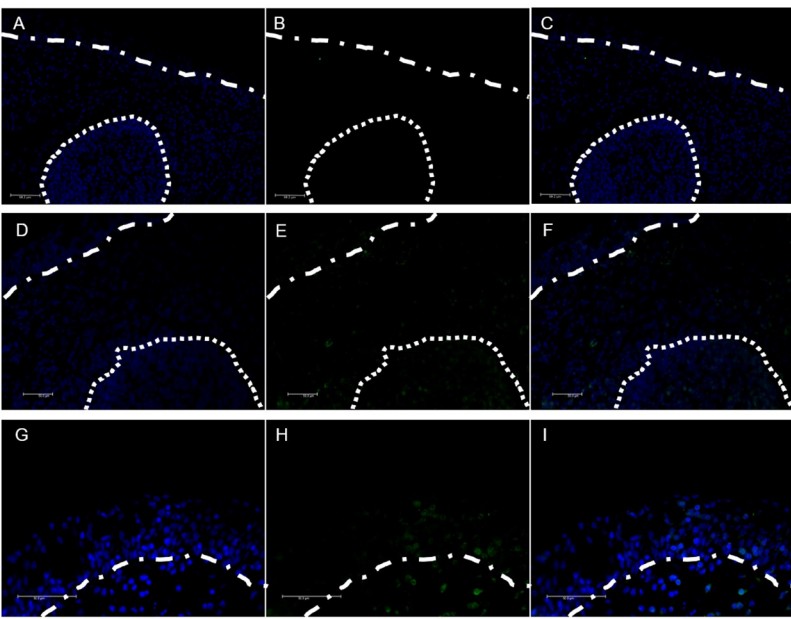

**Fig 11. CD3 immunofluorescence in porcine conjunctiva.** Dotted line marks the limits of the same follicle shown in each of the panels. Dotted-dash lines mark the limit between conjunctival epithelium and stroma. (A) nuclei [blue], (B) Negative control of CD3+ lymphocytes [green], and (C) merged images of (A) and (B). (D) nuclei [blue], (E) CD3+ lymphocytes [green], and (F) merged images of (D) and E). (G) nuclei [blue], (H) CD3+ lymphocytes [green], and (I) merged images of (G) and (H).

also in the conjunctival epithelium. This confirms the presence of intraepithelial lymphocytes in the pig conjunctiva (Fig 11).

## Porcine Meibomian glands

Throughout the tarsal plate of the eyelids, there were many sebaceous Meibomian glands composed of large ducts connected by ductules to numerous acini (Fig 12). The acini were composed of meibocytes in which the cytoplasm was loaded with lipids. In some areas the meibocytes appeared to be disintegrating, freeing the meibum by holocrine secretion into the ductules. The secretion was composed of lipids and meibocyte detritus and forms the lipid layer of the tear film [28].

## Discussion

In this study we analyzed the characteristics of the domestic pig ocular surface, paying special attention to the conjunctiva because it has been studied to a lesser extent than in other potential animal models. Specifically, and due to its role in protecting the ocular surface, we focused on the presence, type, and distribution of conjunctival goblet cells, and on the characteristics of the CALT. In addition, an important objective of this study was to compare the porcine ocular surface with that of humans and determine if the similarities make the pig a good model to study ocular surface pathology and obtain data that can be extrapolated to human eyes.

Although some published studies have reported the characteristics of the porcine eyeball, to the best of our knowledge this is the first time that both the eyeball and eyelids have been analyzed to provide a detailed histological description of the pig ocular surface. The difficulties of obtaining complete eyelids from pigs when the tissue is provided by slaughterhouses may explain the lack of studies.

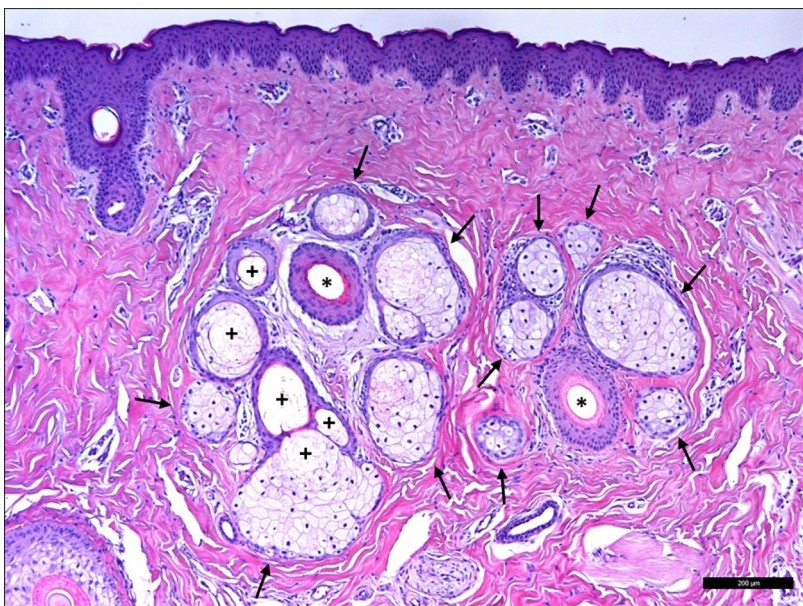

**Fig 12. Meibomian glands in the pig eye.** Different meibocyte acini (arrows) surround the Meibomian gland ductules (asterisks). Areas of meibocyte disintegration were clearly evident (pluses).

A remarkable characteristic of the pig eye is its thick cornea compared to that of humans that measures around 535 μm [29]. We measured corneal thickness in our fixed sections and found that the mean corneal thickness, derived from measurements near the center of the cornea, near the limbus, and at an intermediate position between the center and limbus, was 1,281 ± 71 μm, similar to the 1,248 ± 144 μm that Menduni et al. described in 2018 using an ultrasonic pachymeter [24]. The porcine cornea has an epithelium, stroma, Descemet's membrane, and endothelium, but it lacks Bowman's layer. The presence of Bowman's layer in mammals is controversial. For several years it was thought that only humans and other primates had it. However, more recently, this layer has been described in other animals. For instance, Merindano et al. [30] reported that several herbivores (deer, samba, giraffe, xo, zebu, and eland) have a well-defined Bowman's layer. Although some researchers suggest that pig corneas may have Bowman´s layer, we did not observe that in our study, which agrees with the majority of published reports [30,31]. Therefore, the absence of Bowman's layer and the increased thickness are the main differences of the pig cornea compared to humans. Other than that, the structure is similar, especially regarding the epithelium.

The absence of a nictitating membrane in humans precludes a comparison of this part of the conjunctiva. Apart from that, the porcine conjunctiva is comparable with that of humans. Porcine conjunctival goblet cells occur throughout all the conjunctival epithelium, as in humans. Although some clusters of goblet cells were present in the porcine conjunctiva, the majority of goblet cells were distributed individually, as they are in humans [32]. Thus, in this respect, the porcine conjunctiva would be a better model for normal and pathological conjunctivas than rodents because the rodent goblet cells are mainly grouped in clusters [32].

We found the largest number of goblet cells in the palpebral regions. In humans, the majority of goblet cells are found in the lower eyelid, especially in the nasal region. In the pig conjunctiva, the GCD was also larger in the lower eyelid than in the upper eyelid, but this difference was not statistically significant. In porcine conjunctivas, as in humans, goblet cells were absent in the perilimbal conjunctiva and present in small amounts in the bulbar

conjunctiva. The GCD that we measured in the porcine bulbar conjunctiva, 12.69 ± 4.29 cells/mm, does not differ greatly from the one obtained by Kawano et al. [33] in the same area of humans, 8.24 ± 3.7 cells/mm. We found that 35% of all goblet cells in the anterior ocular surface were located in the nictitating membrane whereas in the nasal bulbar and fornix conjunctiva we found fewer goblet cells than in the central bulbar and fornix conjunctiva. While humans have no nictitating membrane [34], most of the human goblet cells are located in the nasal conjunctiva, the same area where the nictitating membrane is present in pigs.

The goblet cells of the normal human conjunctiva can be labeled with several lectins, such as HPA, PNA, and others. HPA binds specifically to N-acetyl-D-galactosamine residues, of which human goblet cells contain a large amount [35]. In the present study, the pig goblet cell granules also stained strongly with HPA and PNA lectins, and thus have similar residues.

Structured CALT, including follicles and diffuse lymphoid tissue, was abundantly present in the pig conjunctivas, showing the characteristics and topographical distribution described by Knop & Knop in 2000 for humans [9]. The presence of conjunctival follicles in 12 mammalian species, including the pig, was previously demonstrated by Chodosh et al. in 1988 [36]. Interestingly, rodents did not have conjunctival follicles. The relative abundance and distribution of the follicles in our pig conjunctivas was similar to that reported for humans [8,9]. Also, the characteristics of the epithelium that covers the follicles, the FAE, were similar to those described in human conjunctivas, i.e., the absence of goblet cells, thinner epithelium, and discontinuous basement membrane. We also observed a relationship between the number and size of the follicles. The conjunctival sections with the largest number of follicles also had the biggest follicles, suggesting a more developed CALT in these animals compared to others. Perhaps this is related to their habitats, e.g., spending their lives with their heads near or on the ground, although we did not find anything related in the literature. Finally, with anti-CD3 immunostaining, we demonstrated that in addition to the follicles and the diffuse lymphoid tissue, porcine eyes have intraepithelial T lymphocytes in the conjunctival epithelium, as do humans [37].

Based upon our collective observations of the porcine conjunctiva, the main difference between it and the human conjunctiva is the greater surface area of this tissue in pigs due to the presence of the nictitating membrane that is covered by conjunctiva. In humans, the counterpart of the nictitating membrane are the plicae semilunares that exist as folds of the bulbar conjunctiva, connecting it to the caruncle [38]. Their main function is to keep the lacrimal drainage stable with the movements of the eye. Although different, both structures have important similarities, such as the presence of goblet cells and lymphoid follicles. Thus, both help in the lubrication and immune protection of the ocular surface [39].

Tissues for research can be obtained from laboratory animals, among which rats, mice, and rabbits are the most commonly used species in ophthalmology-related studies. For that reason, the majority of the commercially available antibodies show reactivity against these species, whereas just a limited number of them are tested against porcine antigens. However in recent years, the use of porcine eyes has increased. They can be obtained from pigs used in research, but also from slaughterhouses. The reliance on slaughterhouse pigs achieves an ethical benefit in that the animals are not euthanized solely for the purpose of research. Instead the eyes are derived from pigs that are being sacrificed for human consumption under strict regulations and hygienic measures. This fact may have a significant impact on research outcomes by providing more variability than can be obtained with laboratory animals. The increased variability in experimental outcomes may initially seem disadvantageous; however, because higher variability may better represent native, evolutionarily refined processes, it could also be an important advantage in terms of reliability and extrapolation of the results. In addition, another important advantage of using porcine tissues for research is the similarity of pig and human

eye morphology and tear film [24,26], which makes the pig a very useful model to study ocular surface diseases such as dry eye.

A limitation of this study is the lack of information on porcine tears. Unfortunately, obtaining tears from slaughterhouse animals is unfeasible. However, we consider that porcine tear analysis would be of great scientific interest and will try to achieve this in future research. We excluded the lacrimal gland from this study because in 2013 Henker et al. published an exhaustive investigation of the morphology and location of the pig lacrimal gland [26].

Despite the lack of a complete knowledge of the porcine ocular surface, pig eyes have been successfully used to perform functional studies in the ocular surface. For instance, their usefulness in corneal wound healing [40] and in nanoparticle [41] and liposome [42] corneal drug penetration studies has been proved. Now, with the present study, we aimed at promoting the use of this widely available source of healthy biological material to study not just the cornea, but the whole ocular surface.

In conclusion, even though the proteomics and biomechanics may be distinctly different between porcine and human ocular surface tissues, the structural similarities between them, especially the conjunctiva as documented in this study, support the use of pig as a model species for ocular surface disease studies.

## Acknowledgments

Authors thank the personnel from the local slaughterhouse Justino Gutiérrez S.L. (Laguna de Duero, Valladolid, Spain) for kindly providing the pig eyeballs and eyelids used in this study.

## Author Contributions

**Conceptualization:** Yolanda Diebold.

**Formal analysis:** Mario Crespo-Moral.

**Funding acquisition:** Laura García-Posadas, Yolanda Diebold.

**Investigation:** Mario Crespo-Moral, Laura García-Posadas, Antonio López-García.

**Methodology:** Mario Crespo-Moral, Laura García-Posadas, Yolanda Diebold.

**Project administration:** Yolanda Diebold.

**Resources:** Yolanda Diebold.

**Supervision:** Yolanda Diebold.

**Validation:** Mario Crespo-Moral, Laura García-Posadas.

**Visualization:** Mario Crespo-Moral, Laura García-Posadas.

**Writing – original draft:** Mario Crespo-Moral, Laura García-Posadas.

**Writing – review & editing:** Laura García-Posadas, Yolanda Diebold.

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
