## [Decision Letter · Decision Letter 0]

16 Dec 2019

PONE-D-19-32784

Histological and immunohistochemical characterization of the porcine ocular surface

PLOS ONE

Dear Dr. DIEBOLD,

Thank you for submitting your manuscript to PLOS ONE. Your paper has been carefully reviewered by external reviewers. While they found the article is well written and designed, they raised some comments to be addressed. Therefore, we invite you to submit a revised version of the manuscript that addresses the points raised during the review process.

We would appreciate receiving your revised manuscript by Jan 30 2020 11:59PM. To enhance the reproducibility of your results, we recommend that if applicable you deposit your laboratory protocols in protocols.io, where a protocol can be assigned its own identifier (DOI) such that it can be cited independently in the future. For instructions see: http://journals.plos.org/plosone/s/submission-guidelines#loc-laboratory-protocols

We look forward to receiving your revised manuscript.

Kind regards,

Yu-Chi Liu, M.D

Academic Editor

PLOS ONE

Journal Requirements:

2. At this time, we ask that you please provide the source and any product numbers and lot numbers of the lectins PNA and HPA used in your study in the Methods section of the manuscript.

Reviewers' comments:

Reviewer's Responses to Questions

**Comments to the Author**

1. Is the manuscript technically sound, and do the data support the conclusions?

Reviewer #1: Partly

Reviewer #2: Yes

2. Has the statistical analysis been performed appropriately and rigorously? 

Reviewer #1: Yes

Reviewer #2: Yes

3. Have the authors made all data underlying the findings in their manuscript fully available?

Reviewer #1: Yes

Reviewer #2: No

4. Is the manuscript presented in an intelligible fashion and written in standard English?

Reviewer #1: Yes

Reviewer #2: Yes

5. Review Comments to the Author

Reviewer #1: The manuscript characterized the ocular surface of the white domestic pig (Sus scrofa domestica), including cornea, conjunctiva, goblet cell numbers and distribution, mucin composition, CLAT numbers and distribution, lymphoid tissues, Meibominan glands. They discussed the differences and similarities between porcine and human eyes. For example, human eyes do not have nictitating membrane while porcine eyes have; porcine cornea does not have Bowman’s layer and much thicker than human cornea. Despite the differences, porcine conjunctiva is similar to human conjunctiva at goblet cell density and distribution, mucin composition, CLAT and lymphoid tissue characteristics and distribution et al. They conclude that porcine ocular structures are similar to those of humans, is a good model to study ocular surface pathology of human eye.

The manuscript is well written, and the experiments are well designed and analysed. Experiment results are fully discussed and support the conclusion.

Some minor concerns:

1. In line 105, pre-adult pigs (6-8 weeks) were used. Are there any specific reasons to use pre-adult pigs? Will the ocular surface characteristics be different when pigs become adult?

2. Figure legends should not be inserted into the main text.

3. Should include a negative control in Figure 11

Reviewer #2: The article is well written and articulated. There is no major concern with the techniques that they used to answer the research questions. There are only two queries that I hope the authors could address:

1. One of the challenges of using pig tissues for research is the limited number of antibodies that recognize porcine antigens. This perhaps should be highlighted in the discussion.

2. The current study only highlighted the structural similarities between porcine and human corneas and ocular surface. Perhaps this should be made distinct in the conclusion of the study. The proteomics and biomechanics may be distinctly different between the two species.

6. PLOS authors have the option to publish the peer review history of their article (what does this mean?). If published, this will include your full peer review and any attached files.

Reviewer #1: No

Reviewer #2: No

---

## [Author Response · Author response to Decision Letter 0]

24 Dec 2019

RESPONSE TO THE JOURNAL REQUIEREMENTS AND THE SUGGESTIONS OF THE REVIEWERS

Journal Requirements:

Thank you very much for remembering us to accomplish these Journal Requirements.

1. Please ensure that your manuscript meets PLOS ONE’s style requirements, including those for file naming.

We have made the necessary changes in the manuscript to be sure that it follows PLOS ONE’s submission guidelines.

2. At this time, we ask that you please provide the source and any product numbers and lot numbers of the lectins PNA and HPA used in your study in the Methods section of the manuscript.

We have included this information in the Methods section of the manuscript as required.

Reviewer #1:

Thank you very much for your useful comments. As suggested, we have included the negative control in Figure 11. Below we respond to all of your specific questions.

1. In line 105, pre-adult pigs (6-8 weeks) were used. Are there any specific reasons to use pre-adult pigs? Will the ocular surface characteristics be different when pigs become adult?

The reason to use animals in this age (6-8 months, not weeks) is that we use pigs that are designated to human consumption, and that is the age at which they go to the slaughterhouse. We have not found any information in the literature suggesting that the ocular surface characteristics may change after this age. These animals are close to or into their sexual maturation period. We consider that no bigger differences would be found between them and young adult pigs.

2. Figure legends should not be inserted into the main text.

The Plos One Figure instructions demands to “place figure captions in the manuscript text in read order, immediately following the paragraph where the figure is first cited. Do not include captions as part of the figure files or submit them in a separate document.”

3. Should include a negative control in Figure 11.

We have included the negative control of the CD3 inmunofluorescence keeping the structure of the figure: Blue channel (nucleus), green channel (CD3 positive), merge.

Reviewer #2:

Thank you very much for your useful comments and suggestions. Below we respond to all of your specific comments. Changes in the manuscript can be seen in the track changes version.

1. One of the challenges of using pig tissues for research is the limited number of antibodies that recognize porcine antigens. This perhaps should be highlighted in the discussion.

We completely agree with this affirmation and we have now added a sentence in the discussion including this statement.

2. The current study only highlighted the structural similarities between porcine and human corneas and ocular surface. Perhaps this should be made distinct in the conclusions of the study. The proteomics and biomechanics may be distinctly different between the two species.

Thank you for your comment. We have added this in the conclusion to clarify this situation.

---

## [Editor Report · Decision Letter 1]

30 Dec 2019

Histological and immunohistochemical characterization of the porcine ocular surface

PONE-D-19-32784R1

Dear Dr. DIEBOLD,

We are pleased to inform you that your manuscript has been judged scientifically suitable for publication and will be formally accepted for publication once it complies with all outstanding technical requirements.

With kind regards,

Yu-Chi Liu, M.D

Academic Editor

PLOS ONE
---

## [Editor Report · Acceptance letter]

2 Jan 2020

PONE-D-19-32784R1 

Histological and immunohistochemical characterization of the porcine ocular surface 

Dear Dr. DIEBOLD:

I am pleased to inform you that your manuscript has been deemed suitable for publication in PLOS ONE. Congratulations! Your manuscript is now with our production department. 

With kind regards,

on behalf of

Dr. Yu-Chi Liu 

Academic Editor

PLOS ONE